# Network analysis of regional livestock trade in West Africa

**Valerie C. Valerio** [1] *, **Olivier J. Walther** [2], **Marjatta Eilittä** [3], **Brahima Cissé** [4‡],
**Rachata Muneepeerakul** [1‡], **Gregory A. Kiker** [1‡]

**1** Department of Agricultural and Biological Engineering, University of Florida, Gainesville, Florida, United
States of America, **2** Department of Geography, University of Florida, Gainesville, Florida, United States of
America, **3** Cultivating New Frontiers in Agriculture (CNFA), Washington, D.C., United States of America,
**4** Comité permanent Inter-États de Lutte contre la Sécheresse au Sahel (CILSS), Ouagadougou, Burkina
Faso

☉ These authors contributed equally to this work.
‡ These authors also contributed equally to this work.
* valerie.valerio@ufl.edu

**Data Availability Statement:** All relevant data are within the paper and its Supporting Information files.

**Funding:** This work was funded in whole or part by the United States Agency for International

## Abstract

In West Africa, long and complex livestock value chains connect producers mostly in the Sahel with consumption basins in urban areas and the coast. Regional livestock trade is highly informal and, despite recent efforts to understand animal movement patterns in the region, remains largely unrecorded. Using CILSS' database on intraregional livestock trade, we built yearly and overall weighted networks of animal movements between markets. We mapped and characterized the trade networks, identified market communities, key markets and their roles. Additionally, we compared the observed network properties with null-model generated ensembles. Most movements corresponded to cattle, were made by vehicle, and originated in Burkina Faso. We found that live animals in the central and eastern trade basins flow through well-defined, long distance trade corridors where markets tend to trade in a disassortive way with others in their proximity. Modularity-based communities indicated that both national and cross-border trade groups exist. The network's degree and link distributions followed a log-normal or a power-law distribution, and key markets located primarily in urban centers and near borders serve as hubs that give peripheral markets access to the regional network. The null model ensembles could not reproduce the observed higher-level properties, particularly the propinquity and highly negative assortativity, suggesting that other possibly spatial factors shape the structure of regional live animal trade. Our findings support eliminating cross-border impediments and improving the condition of the regional road network, which limit intraregional trade of and contribute to the high prices of food products in West Africa. Although with limitations, our study sheds light on the abstruse structure of regional livestock trade, and the role of trade communities and markets in West Africa.

Development (USAID) Bureau for Food Security under Agreement # AID-OAA-L-15-00003 as part of the Feed the Future Innovation Lab for Livestock Systems. This work was also partly supported by USDA-NIFA NNF proposal #2014-10398 USDA National Institute of Food and Agriculture. Any opinions, findings, conclusions, or recommendations expressed here are those of the authors alone and do not necessarily reflect the view of the United States Agency for International Development (USAID) or the U.S. Department of Agriculture. The funders had no role in study design, data collection and analysis, decision to publish, or preparation of the manuscript.

**Competing interests:** RM (co-author) serves in the Editorial Board of PLOS ONE. However, this does not alter our adherence to PLOS ONE policies on sharing data and materials.

## Introduction

Livestock production involves at least 20 million people across West Africa [1], where long market chains connect producers in the Sahel with consumption basins in urban areas and the coast [2–5]. Livestock production, marketing, and processing generate income for actors along the value chain [1,4] and provide food and nutrition security in the region. Intraregional livestock trade is highly informal; its true magnitude is not captured in official statistics and therefore unknown [6,7]. Livestock (mostly cattle and small ruminants) are traded live and lead the intraregional food trade [8,9].

Despite significant tariff and non-tariff barriers to trade [2,4] and sizeable poultry and beef imports, West African intra-regional livestock flows have increased in the past two decades. The existing gap between production and demand of animal products [3,4,7] is expected to widen in the next decades [10,11] driven by population and income growth, migration and urbanization [3,4], suggesting that there is potential for intraregional livestock trade to satisfy the growing demand for animal products [3]. Moreover, the percentage of intraregional trade in West Africa is considered low when compared to other regions of the world. Because of this, the livestock sector has been recognized as a major agricultural trade opportunity in West Africa, and particularly for the Sahelian countries of Niger, Burkina Faso and Mali [12–14].

The livestock marketing system can be studied as a network of actors or locations connected by animal shipments. Livestock markets are spread throughout the region forming a web with assembling and bifurcating connections through which animals are bought in from rural areas, shipped to wholesalers and then sold in shortage areas, often across borders. Network analysis methods can help gain insights into livestock market structure and functioning, and guide policymakers in designing agricultural development and trade policies. However, network analysis relies on movement data that is often not collected or not available in developing countries.

In West Africa, livestock trade networks appear to be dominated by socially embedded business transactions, generating contrasting opinions on the market system's ability to respond to future increases in demand for livestock-derived products. On the one hand, the market structure is thought to constrain agricultural growth [2,15]; on the other hand, it confers the system with resilience in an environment where disruptions and shocks to the market system–like climatic events and conflict [16]–are frequent [3]. Few attempts to formally describe the regional trade structure have been made to support either claim [17], partly because comprehensive data are lacking on livestock movements, production (allowing inferences on sales made), sales (allowing inferences on domestic vs. regional sales) and exports.

Walther [17–19] used Social Network Analysis (SNA) to study cross-border trade and policy networks in West Africa, but excluded livestock. Three different livestock-specific initiatives have studied animal movements as networks in the region. Dean et al. (2013) studied the risk of disease transmission through cattle trade in Togo. Motta et al. [20], on the other hand, studied the cattle trade network in Cameroon and discussed the implications of the network structure for regional disease spread. More recently, Apolloni et al. [21] and Nicolas et al. [22] mapped, characterized and attempted to predict livestock movements in Mauritania. All these efforts highlighted the importance of international livestock trade; however, all three used data collected in single countries. Given the high percentage of movements and animals that cross borders according to these studies, an understanding of regional patterns of trade is essential to better prepare and respond to disruptions like climatic events (drought), market closures or disease outbreaks that might trigger or worsen famine [16,23]. Although some understanding of animal mobility in the region exists, animal movements originating in the top three exporters (Niger, Mali and Burkina Faso) have not been analyzed as a network.

Our objective is to advance the formal study of regional trade networks in West Africa. We use network analysis to characterize the network of live animal trade (cattle, sheep, goats and donkeys) originating primarily in Burkina Faso, the third largest exporter of live animals in the region [9] as they are shipped through the central and eastern trade basins. We use survey data compiled by the Permanent Interstate Committee for Drought Control in the Sahel (Comité Inter-Etats de Lutte contre la Sécheresse au Sahel, abbreviated as CILSS). The CILSS database covers significantly more livestock trade than official figures, contains records of multiple years (2013–2017) and includes movements from selected corridors between 10 West African countries. We first provide a (1) brief descriptive summary of the movements by type of movement (national or international), livestock and transport. We then (2) map and characterize the market network using network statistics and (3) identify market communities. Lastly, we (4) identify key markets and their roles in the regional network. Our findings are contextualized, and their policy implications are discussed.

## Materials and methods

### Study area

Data used in our study were collected in selected markets and border exit points on livestock corridors throughout West Africa mapped in Fig 1. West Africa extends over 7.9 million km$^2$, occupying approximately one-quarter of the African continent [24]. The region is comprised of 16 countries where hundreds of languages are spoken, 35 of which are spoken by more than 1 million inhabitants [25]. West Africa spans over diverse agroecological zones (arid, semi-arid, sub-humid, humid) and production systems [25], and over largely horizontal stripes of increasing rainfall (<100 to >1600mm) and wet season duration (2–10 months) from north to south [24]. Most of the region experiences unimodal rainfall that allows one growing season from April/June to September/October. Both the Sahel and the Guinean Coast have experienced drier conditions since the late 1960s [26,27] which, combined with severe recurring drought periods, have affected agricultural production. The agriculture sector concentrates more than half of each country's GDP, while the livestock sector contributes 5–44% of the agricultural GDP [4], carrying greater importance in Sahelian countries where most of the region's 220 million ruminants are located [25,28]. In contrast, there are 362 million humans in West Africa (est. 2016)– 46% residing in cities [29]–the majority of which concentrate in coastal countries [25].

### Data source and pre-processing

Data were collected between January 2013 and August 2017 by the CILSS and regional partner organizations to track the direction and magnitude of intraregional trade. The data collection methodology is described in S1 File. Survey locations include markets and border crossings in selected trade corridors connecting Benin, Burkina Faso, Côte d'Ivoire, Ghana, Mali, Niger, Nigeria, Senegal and Togo. The "markets" included in our study, mapped in Fig 1, are defined as geographical locations that were either (i) survey points or (ii) reported as an origins or destinations of animal movements.

Although it does not capture all intraregional livestock trade, Josserand [6] estimated that CILSS's livestock trade database captured significantly more trade than official figures, especially for animals originating primarily in Burkina Faso and Mali, two of the three biggest livestock exporters in the region (after Niger) [9]. However, the subset of movements used in our analysis primarily captures movements departing Burkina Faso.

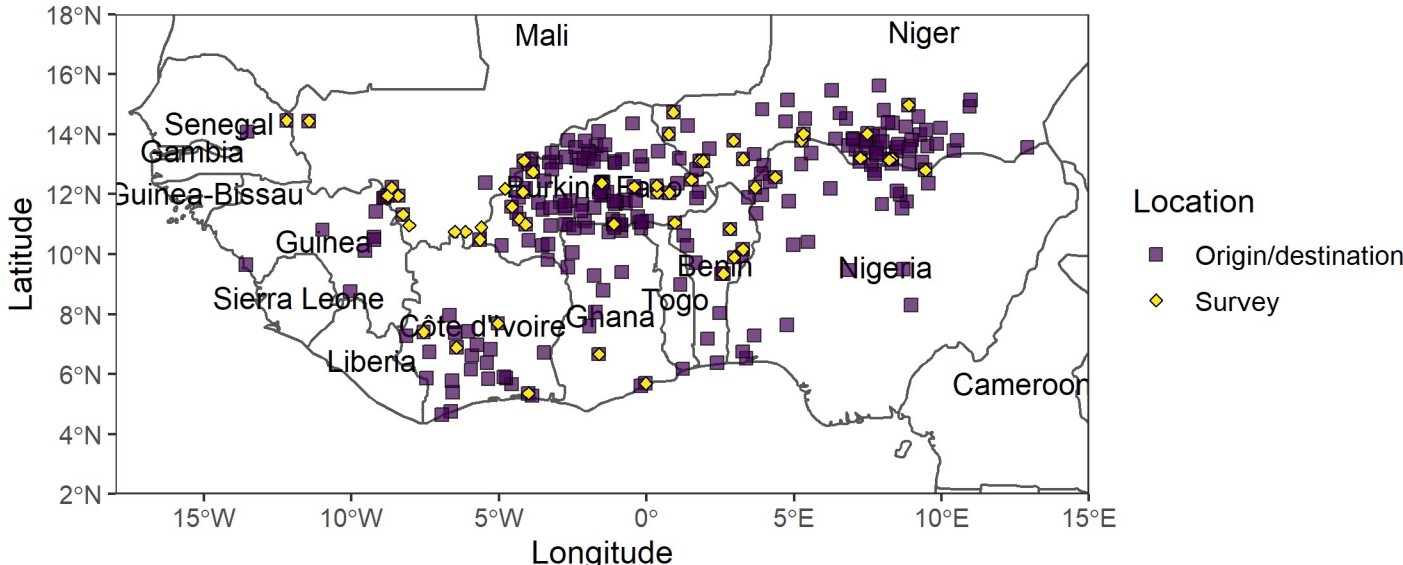

**Fig 1. Map of survey, origin and destination markets recorded in the livestock trade database from 2013–2017.** Diamonds indicate livestock markets where data was collected or survey markets; squares highlight reported origin and destination markets. The map was made in R with the sf package [30].

Collected information included the date of each movement, the type of livestock (goat, sheep, cattle or donkey), volumes, origin and destination market and country, and type of transport (on the hoof, by truck or train). These variables are defined in Table 1.

Data were pre-processed in Microsoft Excel and R Statistical Software [31] to homogenize and code the variable values. In the database, each entry represents a livestock movement from a loading point to an unloading point, passing through a data collection point (or origin, destination and survey markets, respectively). We separated each entry into two movements: one from the origin market to the survey market, and one from the survey market to destination market. To avoid self-loops in the network, entries were only separated if their origin or destination markets differed from the data collection point. Some of the movements were missing either the origin, destination or both markets, while other shipments could not be geolocated. Excluding incomplete and unmapped entries resulted in 42,252 movements.

**Table 1. Variables from CILSS' database on intraregional livestock trade used in our analysis.**

| Name | Variable[a] | Definition | Format |
|---|---|---|---|
| Loading point | from (name) | Origin location of the livestock movement | Code |
| Unloading point | to (name) | Destination location of the livestock movement | Code |
| Origin country [b c] | fromc (country/cid) | Country where the origin market is located | Name/Code |
| Destination country[b c] | toc (country/cid) | Country where the destination market is located | Name/Code |
| Date | dmonth | Month the movement was recorded | Month |
| Type | type | Type of livestock (goat, sheep, cattle or donkey) | Category |
| Heads | weight | Scaled number of animals being transported | Number |
| Type of transport | transp | Type of transport used (vehicle, on foot or by train) | Category |
| Type of movement[c] | intl | Indicates if the movement crossed borders | Category |

[a]The "variable" column contains the names of the variables provided in S2 and S3 Files. The variable names in the node list are provided in parentheses.

[b]Countries where the origin/destination markets were located were used. In some cases, these differed from the countries recorded in the database.

[c]Computed by authors

Because of the geographic scale at which the movements were recorded, the shipment weights have been scaled to protect the identity of individual traders and/or shipments. This scaling does not affect the construction or analysis of the network. Data comprising complete mapped entries are provided in S2 and S3 Files as an edge list and its corresponding node list. The data processing pipeline is shown in Fig 2.

**Network construction.** The network was constructed by representing markets and shipments between markets as nodes and directed links, respectively. Links were weighted by the (scaled) number of animals involved in each movement, or by the number of movements between the markets. The movements were aggregated over each year (2013–2017), resulting in 5 directed static networks with weighted links. Additionally, we constructed one overall network (2013–2017) to study its degree distribution.

## Descriptive summary

We provide a descriptive summary of the data. This serves two purposes: describing the working dataset and allowing for comparisons with previous work. The descriptive summary includes disaggregation of number of movements by type of movement (national or international), livestock (cattle, goat, sheep or donkey), and transport (vehicle or on foot). Additionally, we summarize the shipments by their origin and destination countries.

## Network analysis

We characterized the size, connectivity, heterogeneity and centrality of the trade network using the metrics shown in Table 2. Network-level metrics included the number of data collection points, markets, shipments and pairs of trading markets, the network diameter, link density, average link degree, average shipments, transitivity, average path length, the edge, betweenness and closeness centralization, transitivity, and propinquity.

We expect animals to flow from their origin through local markets of increasing importance before reaching a small number of regional trade hubs and subsequently, consumers. Thus, we hypothesize that the geographic distances between pairs of locations that don't trade are larger than the distances between those that do. To test our hypothesis, we ran a one-sided Mann-Whitney test between the Haversine distances separating markets that did trade animals and the distances between markets that did not ($H_0$ = The location shift between the distances separating markets that did not trade and those that did equals zero; $H_1$ = The location shift between the distances is greater than zero). The Haversine distances between each pair of markets in the yearly networks were calculated using their geographic coordinates with the geosphere R package [32]. A p-value of the one-sided Mann-Whitney test is reported per year.

It is common practice to compare empirical network to a null model ensemble to distinguish important network properties from trivial ones, and to enable comparisons with other networks [33]. Here, we follow the process described in [34] to normalize the metrics of the observed networks. First, we generated 5 configuration model ensembles (one for each year) without self-loops or multiple edges and with 10l re-wiring iterations. Then, network-level metrics were calculated for the 1000 simulated networks in each ensemble. Using the configuration model implies that some properties will not vary between the observed and simulated networks; thus, the number of survey markets, markets, shipments and links (total and average), the link density, and degree and closeness centralities were not computed, while the minimum ensemble p-values of the one-sided Wilcox test are reported for the propinquity. Finally, we compared how the observed properties deviated from the ensemble by calculating a z score for each observed metric per year for the rest of the statistics.

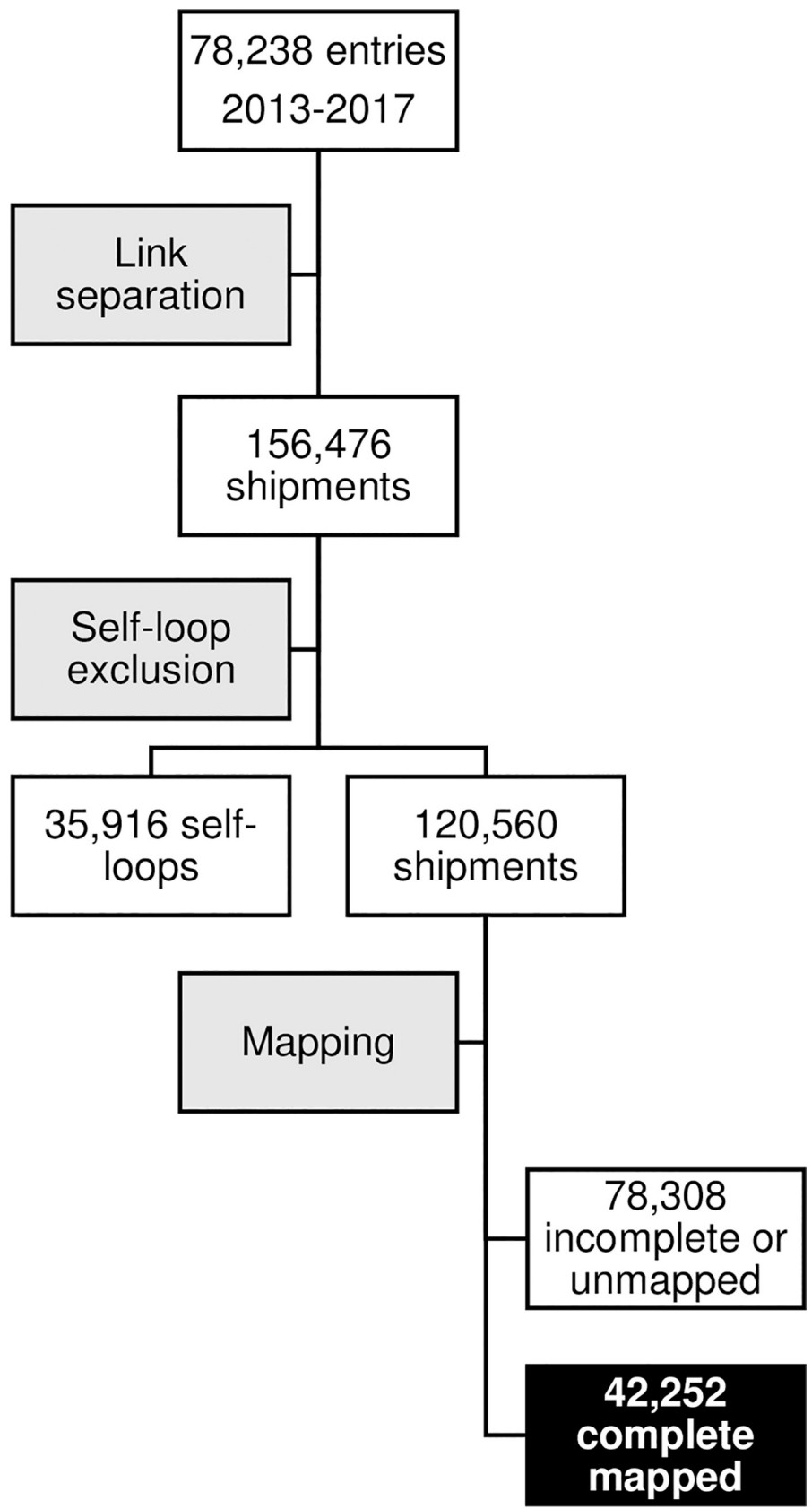

**Fig 2. Data pre-processing pipeline.** Processing steps are shown in grey and data in white. Pre-processing included separating movements into two links, excluding self-loops and incomplete and unmapped entries. After pre-processing, a working dataset of 42,252 movements remained (shown in black).

Additionally, we tested if the degree distribution of the market network aggregated over the study period (2013–2017) follows a power law or a log-normal distribution following Clauset et al. [35] using (a) the number of movements and (b) the number of neighbors as degrees. At the subset level, we calculated the size of the Giant Strongly and Weakly Connected Components (GSCC and GWCC), and detected trade communities with the fast greedy modularity optimization algorithm introduced in [36]. Key nodes were identified with their trade volume and degree (all, in, out) [37]. Market communities and market level metrics were calculated for the most recent year only (2017). The networks were constructed, analyzed, simulated and visualized with the igraph package [38] in R [31]. Additional R packages used to map or analyze the animal movements are specified under each Fig or Table. All the statistics are defined in Table 3.

## Results

### Descriptive data summary

**Type of movement, livestock and transport.** Close to two thirds of the livestock movements were international, while the rest did not cross borders. Table 4 shows that most movements corresponded to cattle, followed by sheep and goats with an insignificant number of donkey shipments. Movement peaks, shown in Fig 3, occurred in the months preceding Tabaski (or Eid al-Adha) some of the years, particularly for sheep. Tabaski, also known as the "festival of the sacrifice" is a worldwide Islamic holiday that affects animal mobility patterns in West Africa because many Muslim families sacrifice sheep to commemorate it. This festival is observed on the lunar calendar and is celebrated on a different Gregorian date (about two

**Table 2. SNA metrics used to characterize the livestock trade network.**

| Type of metric | Network | Subset | Node |
|---|---|---|---|
| **Size** | Survey markets | | Volume (weight) |
| | Markets | | |
| | Shipments | | |
| | Pairs of trading markets | | |
| | Diameter | | |
| **Connectivity/Cohesiveness** | Link Density | GSCC | |
| | Average Link degree | GWCC | |
| | Average shipments | | |
| | Transitivity | | |
| | Average Path Length | | |
| **Heterogeneity** | Propinquity | Trade communities | |
| | Degree distribution | | |
| | Degree assortativity | | |
| **Centrality (unweighted)** | In degree | | Degree |
| | Out degree | | In degree |
| | Betweenness | | Out degree |
| | Closeness | | |

**Table 3. Network metric definitions in the context of livestock trade.**

| Metric | Definition[a] |
|---|---|
| Survey markets | Number of markets where data collection took place. |
| Markets | Number of markets (nodes) that were origins or destinations of livestock shipments (n). |
| Shipments | Number of shipments between markets (m); includes all individual shipments made between all pairs of markets and is different from pairs of trading markets (l). |
| Links (Pairs of trading markets) | Pairs of markets that traded at least one animal (l); directed link (e.g. shipment from market A→B is different than from B→A). |
| Diameter | The longest geodesic distance between any pair of livestock markets in the network using the shortest possible walk from one market to another [39]; calculated considering (directed) and neglecting (undirected) link directions. |
| Link Density | Ratio of links (l) among livestock markets (n) in the network with respect to the maximum possible number of links (2n(n-1)); defined as l/2n(n-1) [39]. |
| Average link degree | Average number of markets that each market traded with; defined as l/n. |
| Average shipments | Average number of shipments each market is involved in; defined as m/n. |
| Transitivity (clustering coefficient) | If we define the neighbors of a specific market as the other markets who are directly linked to it, the clustering coefficient measures the proportion of neighbors of a specific market that are linked to each other (at the node level), or the average of these local clustering coefficients (at the network level) [39,40]. |
| Average path length | The geodesic (shortest path) between two livestock markets averaged over all pairs of livestock markets in the network; defined as $1/n(n-1) \sum_{i \neq j} d(v_i, v_j)$ where $d(v_i, v_j)$ is the geodesic path between markets i and j. |
| Propinquity | The tendency of trading markets to be closer than markets that don't trade. Measured with the p-value of a one-sided Mann-Whitney test between two groups of geographic distances: the distances between pairs of markets that are linked and those between pairs markets that are not linked (or between pairs of markets that traded and those that didn't trade). |
| Degree distribution | Probability distribution of the number of neighbors of each market over the whole network and study period. |
| Degree assortativity | Correlation between the degrees of linked markets, quantifying the tendency of markets to connect with other similar markets in terms of degree centrality (or number of neighbors). |
| Degree centrality; centralization | Number of markets a specific market is connected to; standardized mean difference between degree centrality of the most central market and the rest of the markets [37] (in- and out-degree refers to the number of markets that ship livestock to a market of interest, and the number of markets that the market of interest sends livestock to, respectively [39]). |
| Closeness centrality; centralization | Number of markets a specific market is connected to; standardized mean difference between closeness centrality of the most central market and the rest of the markets [37]. |
| Betweenness centrality; centralization | The frequency a market lies in the shortest path between pairs of markets in the network [37]; Standardized mean difference between betweenness centrality of the most central market and the rest of the markets [39] |
| Giant Strong Connected Component (GSCC) | Maximum connected subset of markets in the network in which all pairs of markets are linked, considering the direction of the links [39]. |
| Giant Weak Connected Component (GWCC) | Maximum connected subset of markets in the network in which all pairs of markets are linked, neglecting the direction of the links [39]. |
| Trade communities | Market community configuration that maximizes the modularity Q or the difference between the links running within communities and those expected by chance [41]. Calculated with the fast greedy algorithm as introduced by [36]. |
| Volume | Livestock volume received or sent by a market. |

[a]Metric sources are cited, definitions adapted to livestock networks were partly drawn from Dubé et al. [42].

**Table 4. Summary of livestock movements for 2013–2017 by type of movement, transport and livestock in absolute and relative quantities.**

|  | Number of movements | Percentage of movements |
|---|---|---|
| **Total** | **42,252** | **100%** |
| International | 24,974 | 58% |
| National | 17,278 | 42% |
| On the hoof | 1,870 | 4% |
| Train | 498 | 1% |
| Vehicle | 39,884 | 94% |
| Cattle | 31,080 | 74% |
| Donkey | 1 | <1% |
| Goat | 2,546 | 6% |
| Sheep | 8,625 | 20% |

weeks earlier) each year. Vehicles (train and truck) were the main form of transport for all livestock types and seasons, as S1 Table and S3 Fig show.

**Origin and destination countries.** Fig 4 shows the origin and destination countries of the shipments. Most movements originated in Burkina Faso, Côte d'Ivoire, and Ghana, with smaller proportions departing Benin, Niger and Mali. Close to a quarter of shipments were destined to Ghana, Côte d'Ivoire and Benin each, whereas Nigeria was reported as the destination country for close to a twentieth of all transfers. Less than 5% of the shipments were leaving to Niger, Mali, Guinea and Senegal combined.

## Network analysis

**Network level: Well-defined long-distance corridors.** Links for each year are mapped in Fig 5 by type of movement (national or cross-border). A line connecting two markets (squares) indicates that at least one animal movement was recorded between them in that year. There are clear differences between the years, yet some movement patterns persist. For example, in 2014 and 2017 there were more pairs of trading markets than other years, as well as more international connections. In 2017, there were Nigerien shipments towards the Nigerian border that were not captured in other years. On the other hand, a general north-south direction of animal movements persists all years. Repeating national patterns also include movements from North-Western Burkina Faso towards Benin and Togo, shipments from central Côte d'Ivoire towards markets in the south and west, and national flows from Northern to Southern Ghana. International movements that persist were made from Burkina Faso into the Ivorian and Ghanaian coasts.

Table 5 shows temporal changes in the data collection during the study period. For instance, the number of survey markets fluctuated reaching their maximum in 2017. The number of shipments peaked in 2014, even when the duration of data collection is considered (full years vs. Jan-Aug for 2017). However, more pairs of trading markets were recorded for 2017. On average, each market traded with 1–2 other markets every 4–10 days each year. Less than 12% of every market's neighbors were connected to each other, and the shortest distance between any two markets ranged between 1–3 links. The propinquity p-values in Table 5 suggests that markets that traded were significantly closer to each other than those that did not for all years except 2014. On the other hand, a negative correlation between the degrees (disassortativity) of markets that traded was found for all years, indicating that markets tend to trade with markets of dissimilar centrality. The centrality values show that some markets trade with more locations than most markets in the network, or that there are hub markets.

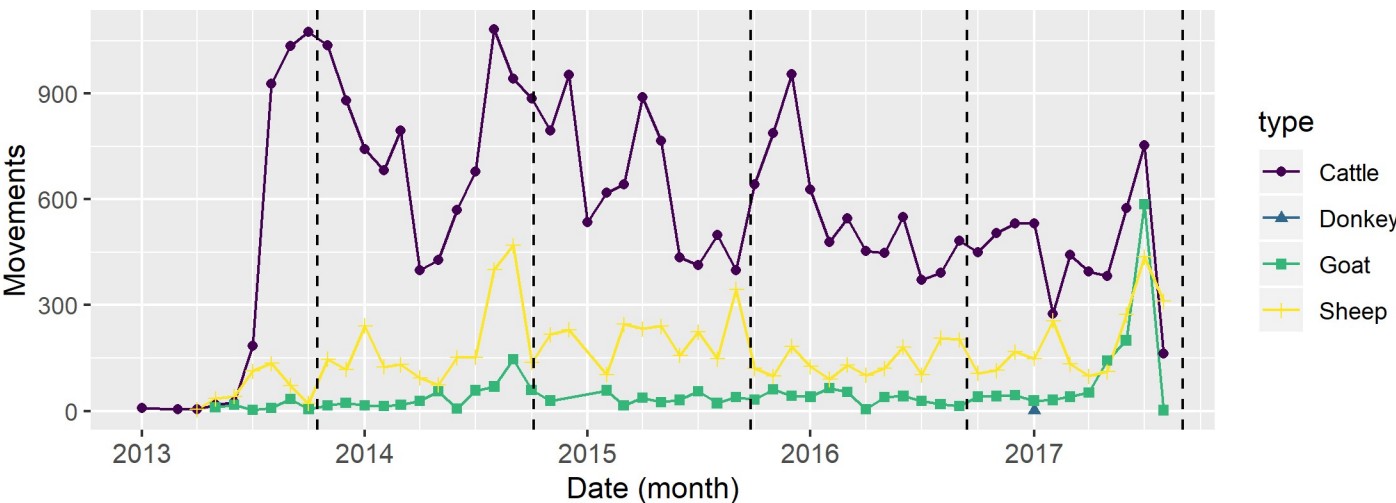

**Fig 3. Number of livestock movements by livestock type and month 2013–2017 for our working dataset.** Sheep movements increase significantly in the months preceding Tabaski, while cattle and other livestock shipments seem to be unchanged by its occurrence. Vertical black dashed lines indicate the occurrence of Tabaski each year. Major ticks (labeled) correspond to the start of each year, whereas minor ticks are quarters (3-month periods).

The z scores of the observed metric values are reported in Table 6. For all years, the degree assortativity and betweenness centrality scores were negative, while the undirected diameter scores were positive. The observed assortativities, however, were 9 or more standard deviations lower than the mean ensemble assortativities. The remaining metrics (directed diameter, transitivity average path length) either had smaller absolute z scores or had both positive and negative scores. Of these, the directed diameter and average path length reached their maximum normalized score in 2017. For all years, the propinquity of the simulated networks was insignificant as the minimum ensemble p-values were all close to 1.

We found that the degree distribution of the network follows either a power-law or a log normal distribution. For the movement distribution shown in Fig 6A, the hypothesis p-value ruled out a log-normal as a plausible fit. However, both a power and a log-normal distribution

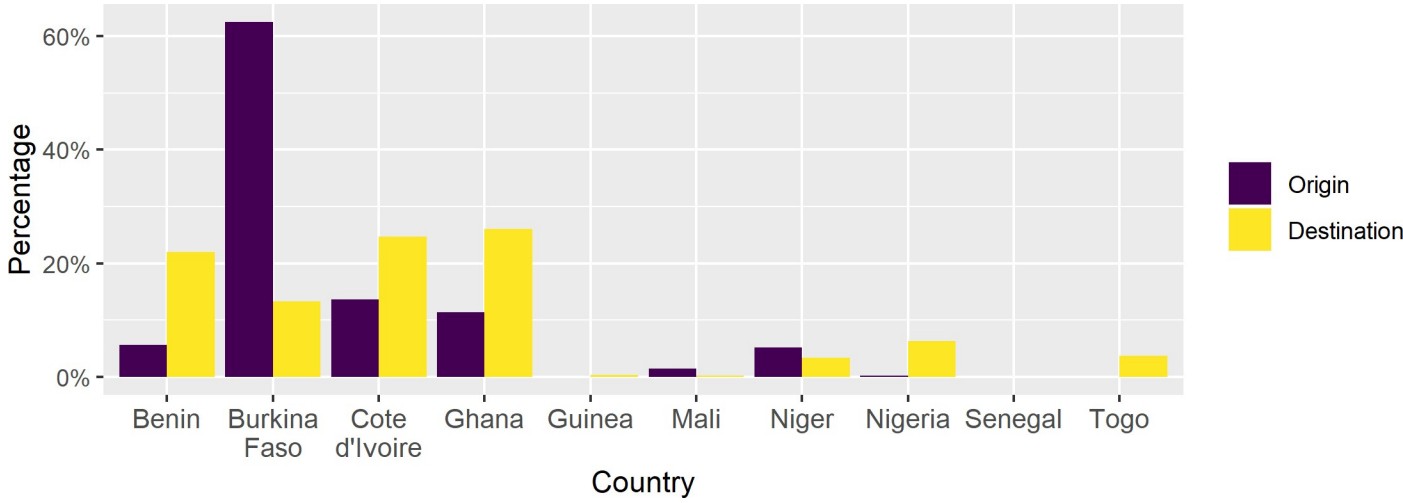

**Fig 4. Proportion of livestock movements by origin and destination country.** The percentage of all movements that originated and were destined to each country are shown in purple and yellow, respectively.

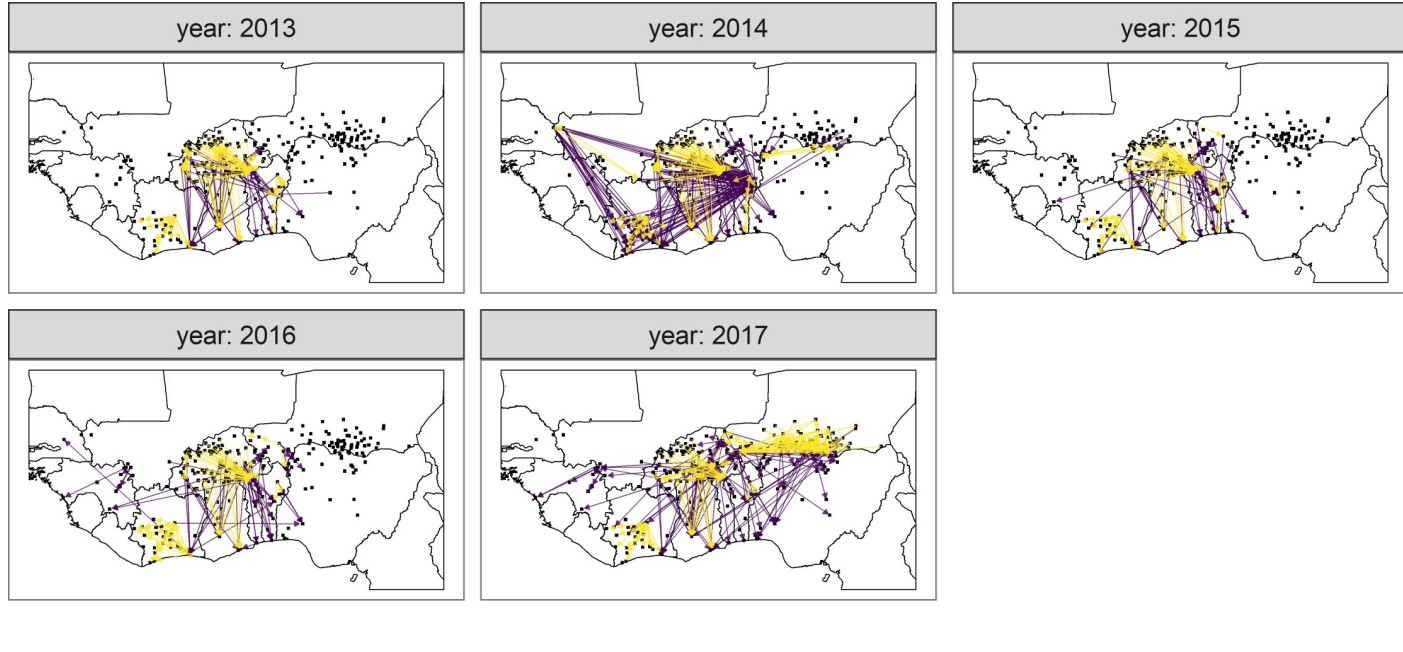

**Fig 5. Livestock shipments by year.** The color of the link indicates if the movement crossed borders (purple) or not (yellow), while black squares mark the origin/destination markets. The maps were made in R by cropping OCHA ROWCA's administrative level-0 boundaries for West and Central Africa. OCHA ROWCA's maps are protected under the CC BY license (https://creativecommons.org/licenses/by/4.0/legalcode).

were plausible for the link (neighbor) distribution shown in Fig 6B. While Vuong's test results strengthened the case for the power law over the log-normal our sample was too small to significantly rule out either one (R = -0.515, $x_{min}$ = 5, p = 0.697, see S2 Table for details).

**Subset level: Cross-border trade communities.** Few markets were reachable by all the other markets via directed paths, as shown by the size of the Giant Strongly Connected Components in Table 7. When the direction of the links was neglected, all the markets belonged of the Giant Component in the yearly networks except for 2016. Markets were classified into 8 groups or trade communities with the fast, greedy modularity optimization algorithm (Q = 62.7%).

Fig 7 shows the 2017 trade network by country and trade community membership. Country membership (Fig 7A and 7C) suggests that many peripheral markets tend to connect to other higher-centrality markets in their country. In turn, these hub markets serve as bridges between different country-communities, and either primarily send or receive animals to/from many other markets. However, different communities emerge when the network is partitioned to maximize the modularity. Some of those communities are essentially single-country groups–like communities 4 and 6 in Burkina Faso and Cote d'Ivoire. Others spread across national boundaries. Cross-border communities include two between Niger and Nigeria (communities 1 and 5), and one extending from Southwestern Niger to Burkina and south towards the coast (2). Some of these groups serve as trade funnels (1) while others are diffusers (6). Both community partitions highlight the relevance of cross-border ties and suggest that markets play distinct structural roles in the network.

**Market level: Border and urban hubs.** Nine key markets, shown in Table 8, were identified with the node level metric values. Key markets were either border markets (within 50 km of an international border), in large urban centers or export markets. Key border markets

**Table 5. Network level metrics for the livestock market network by year.**

| Year | 2013 | 2014 | 2015 | 2016 | 2017[a] |
|---|---|---|---|---|---|
| Survey markets | 25 | 30 | 23 | 33 | 41 |
| Markets | 112 | 136 | 108 | 122 | 183 |
| Shipments | 5997 | 11872 | 10098 | 7914 | 6371 |
| Pairs of trading markets | 154 | 286 | 138 | 146 | 298 |
| Diameter (directed) | 3 | 4 | 5 | 4 | 8 |
| Diameter (undirected) | 7 | 7 | 8 | 7 | 8 |
| Link Density | 1.2% | 1.6% | 1.2% | 1.0% | 0.9% |
| Average Link Degree | 1.4 | 2.1 | 1.3 | 1.2 | 1.6 |
| Average Shipments | 53.5 | 87.3 | 93.5 | 64.9 | 34.8 |
| Transitivity (clustering coefficient) | 4.5% | 11.6% | 2.5% | 2.7% | 7.4% |
| Average Path Length | 1.6 | 1.9 | 1.4 | 1.7 | 3.3 |
| Propinquity (p-value) | 7.58E-05*** | 0.17[ns] | 2.94E-05*** | 4.06E-14*** | 3.69E-60*** |
| Degree Assortativity | -59.9% | -60.9% | -64.0% | -54.8% | -55.2% |
| Degree centralization | 17.8% | 20.4% | 21.9% | 16.5% | 10.7% |
| In-degree centralization | 35.7% | 41.4% | 43.7% | 32.1% | 16.1% |
| Out-degree centralization | 13.2% | 14.0% | 13.8% | 14.7% | 10.6% |
| Betweenness centralization | 0.1% | 0.3% | 0.3% | 0.4% | 0.5% |
| Closeness centralization[b] | 0.6% | 2.2% | 0.4% | 0.6% | 3.1% |

[a] Data for 2017 include movements from January-August

[b] Not well-defined for disconnected graphs

*** p-value<0.001

[ns] Not significant

dominate the in-degrees and include Nadiagou (Burkina Faso, Benin), Dan Barto (Niger, Nigeria) and Dèrassi (Benin, Nigeria). Out-degree leaders are predominantly big urban and export markets, that also serve as hubs but connecting international production to consumer markets in urbanized areas and the coast. Port-Bouët (Côte d'Ivoire), Bouaké (Côte d'Ivoire), Kumasi (Ghana) and Bobo-Dioulasso (Burkina Faso) are located in or near urban settlements. Although close to a border, we also consider Parakou (in Benin, bordering Nigeria) as an urban market because it is in an important urban settlement. The remaining key market is Fada N'Gourma (Burkina Faso) through which livestock mainly destined for export flows.

**Table 6. Z scores of the observed metric values and propinquity significance of the ensembles.** Observed metrics were normalized using the average and standard deviations of the corresponding ensemble values for each year.

| Year | 2013 | 2014 | 2015 | 2016 | 2017 |
|---|---|---|---|---|---|
| Diameter (directed) | -1.7 | -0.8 | 0.2 | -0.8 | 2.8 |
| Diameter (undirected) | 1.3 | 1.3 | 2.8 | 1.3 | 2.8 |
| Transitivity | -2.4 | 4.7 | -4.4 | -4.2 | 0.6 |
| Average path length | -0.7 | -0.1 | -1.2 | -0.6 | 2.8 |
| Degree Assortativity | -11.6 | -12.1 | -13.7 | -9.0 | -9.2 |
| Betweenness centralization | -1.0 | -0.8 | -0.8 | -0.7 | -0.7 |
| Propinquity (min p-value)[a] | 0.99[ns] | 0.99[ns] | 1.00[ns] | 1.00[ns] | 0.97[ns] |

[a] The minimum ensemble propinquity p-value is reported for each year

[ns] Not significant

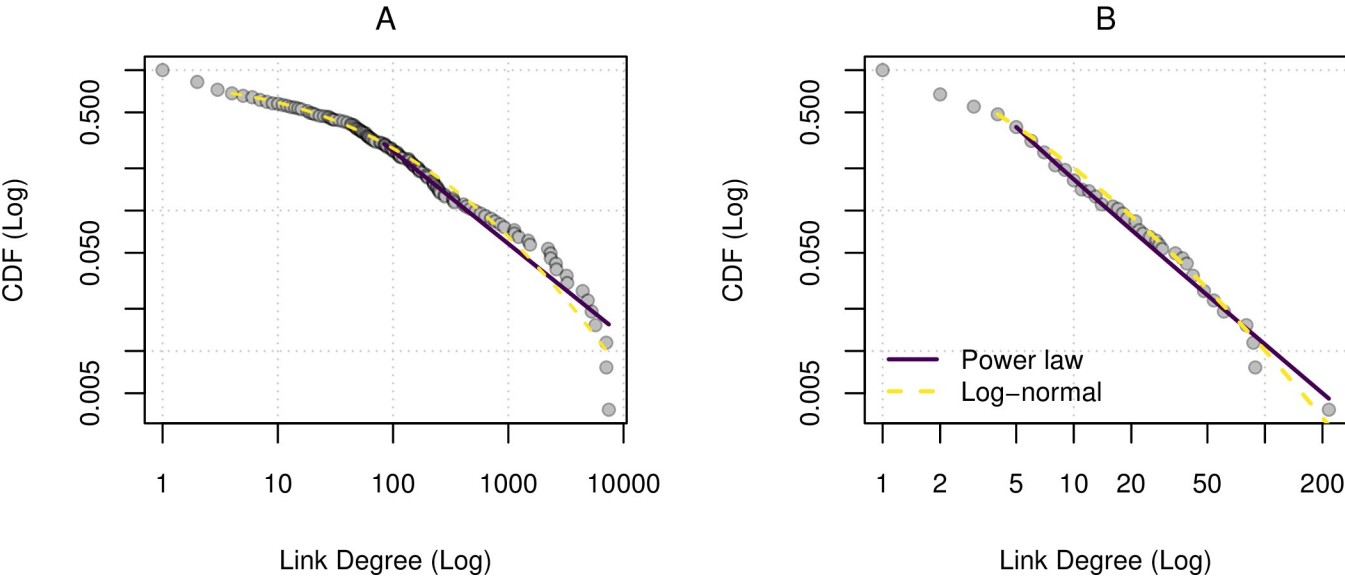

**Fig 6. Power-law fit to aggregated network degree distribution.** Power law (solid line) and a log-normal distribution (dashed line) were fit to the degree distribution of the trade network (n = 262) over the whole study period using the sequences for number of movements and aggregated links. (A) A power law distribution provided a plausible fit to the movement sequence ($x_{min}$ = 84, p = 0.4, $\alpha$ = 1.659). (B) Both a power law ($x_{min}$ = 5, p = 0.69, $\alpha$ = 2.152) and a log-normal ($x_{min}$ = 4, p = 0.45, $\mu$ = -2.410, $\sigma^2$ = 0.036) distribution were plausible for the link sequence, and one could not be favored over the other ($x_{min}$ = 5, R = -0.961, p-value = 0.697). See S2 Table for details.

## Limitations

As in Thébaud et al [1], we do not report the traded volumes for various reasons, including the possibility that some animals and/or shipments were accounted for more than once. Two other main limitations of our study pertain the data: its completeness and changes in the data collection. Incomplete and/or imperfect data are pervasive in the developing world, where the mechanisms to thoroughly capture, store, analyze and communicate findings on movements of goods are often not in place. In our case, incomplete entries could not be used in our analysis and thus limited it. Furthermore, survey markets were selected based on a regional assessment of trade and do not constitute a random sample of all markets, so our work does not paint a complete or unbiased picture of trade patterns in the region but is focused on animals originating in Burkina Faso and Mali that cross international borders in the central and eastern basins of trade of West Africa.

There are two main reasons to be cautious when comparing our results with related research. First, observed temporal fluctuations in network structure could reflect the changes in the data collection and incomplete entries, and not real changes in the movement structure. Movement data was collected by the CILSS to track the direction and magnitude of intraregional livestock trade, and not specifically to carry out this analysis. The second reason pertains

**Table 7. Size of the connected components.**

|  | 2013 | 2014 | 2015 | 2016 | 2017 |
|---|---|---|---|---|---|
| Giant Strongly Connected Component size (markets) | 5 | 4 | 1 | 2 | 12 |
| % of markets in GSCC | 4.5% | 3.7% | <0.1% | 0.2% | 9.8% |
| Giant Weakly Connected Component size (markets) | 112 | 136 | 108 | 105 | 183 |
| % of markets in GWCC | 100.0% | 100% | 100.0% | 86.1% | 100.0% |

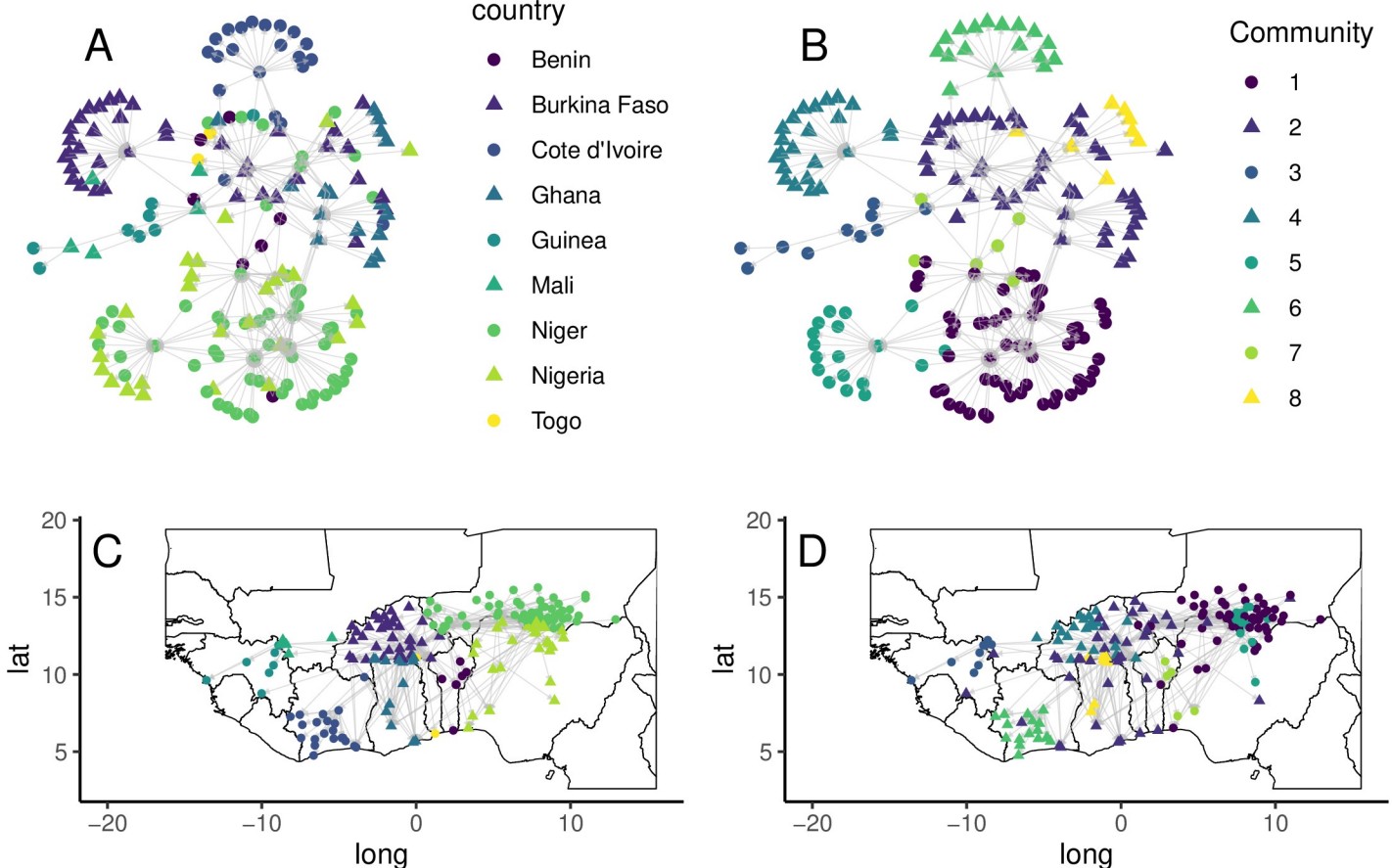

**Fig 7. Market network by country and trade communities for 2017.** Trade communities were detected using igraph's implementation of the fast greedy [36] modularity optimization algorithm (Q = 62.7%, n = 8). Node shapes and colors indicate the country (A, C) or community (B, D) the node belongs to.

related work [20–23] having different purpose, geographic coverage and scale, and using different data collection and analysis methods. Because of these differences, it is possible existing research captured transhumance and/or trade-related, formal or informal movements, or different sections of the value chain. Even if identical methods had been used to investigate animal mobility in previous work, networks of different size, density and/or structure should not

**Table 8. Key nodes in the market trade networks by year and metric.**

| Year | | 2013 | 2014 | 2015 | 2016 | 2017 |
|---|---|---|---|---|---|---|
| Volume | | *Parakou* | *Port-Bouët* | *Port-Bouët* | *Bouaké* | *Bouaké* |
| Links | Degree | Nadiagou | Nadiagou | Nadiagou | Nadiagou | Dan Barto |
| | In-deg. | Nadiagou | Nadiagou | Nadiagou | Nadiagou | Dan Barto |
| | Out-deg. | *Bouaké* | Derassi | *Bouaké* | *Bouaké* | **Fada N'Gourma** |
| Shipments | Degree | *Parakou* | Nadiagou | Nadiagou | *Kumasi* | Dan Barto |
| | In-deg. | *Parakou* | *Parakou* | *Parakou* | *Kumasi* | *Port-Bouët* |
| | Out-deg. | *Bobo-Dioulasso* | Nadiagou | **Fada N'Gourma** | *Bobo-Dioulasso* | *Bouaké* |

Markets with the highest total volume, and total, in and out degree in number of links and shipments are reported for each year. Border markets are markets within 50km of an international border. Urban markets are highlighted in *italic*. Fada N'Gourma, an export market, is in **bold**.

be directly compared. Recent developments in network similarity and distance measures can be used to compare networks of different characteristics, but such a comparison is out of the scope of our work. To enable comparisons with future animal mobility network research, we normalized the observed statistics using simulated network ensembles for each year.

## Discussion

Live animals are the most valued food product traded within West Africa; however, a considerable part of this trade is unofficial [6]. As trade liberalization continues to be pursued in the region, understanding spatiotemporal patterns of regional, live animal movements will become increasingly important to inform policies that support existing and future trade, as well as strategies for disease surveillance and control. We built weighted networks of animal movements originating primarily in Burkina Faso by representing locations (markets) as nodes and shipments as directed links between nodes. We then mapped and characterized the network, identified market communities, key markets and their roles.

Our findings show that in the central and eastern trade basins of West Africa, most animal movements crossed borders, and that those movements were made primarily by vehicle on well-established long-distance routes. The existence of major trade routes is supported by two findings: markets traded with, on average, few other markets multiple times each month, and repeating spatial patterns of trade despite changes in the data collection–like a general north-south movement and defined country roles. Despite these general patterns, we found heterogeneity in the roles that markets play in the network. For instance, markets that traded at least once were significantly closer to each other that those that did not for most years. Combined with the heterogeneous distribution of the number of shipment and trade partners, this indicates that there are hub markets that tend to trade with nearby peripheral ones, and that those hubs connect said periphery to the rest of the network.

The degree distribution followed a power law, which may suggest that the network is more vulnerable to targeted attacks or interventions than to random ones [43]; however, implications of unrecorded and/or incomplete links must be studied further for this specific type of network to assert this. We attempted to reconstruct the observed networks from a null model but local constraints (the degree or link sequences) were not able to reproduce some of the observed network properties. The empirical networks were considerably more disassortative and propinquital than the ensembles, which indicates that other (possibly spatial) processes are necessary to explain higher-order properties of the network. Our findings reflect the fragmentation of the road network inherited from the colonial period and the poor accessibility of many peripheral markets in the region. Cattle trade relies on a handful of paved roads in each country and on a limited number of transnational routes that help connect the Sahel to the main consumption centers [49].

Temporal variation in the network structure from year to year can reflect (not exclusively) changes in the data collection, climatic conditions of historical events, or a combination of them. Untangling their individual contributions is a complex task that we do not attempt here; however, we provide some context that can inform observed changes in the network. This context is provided while considering that our dataset consisted in a subset of all the shipments recorded at a sample of market locations, and that this sample changed during the study period. In 2015, a late start to the rainy season and unfavorable pasture conditions for the second year in a row triggered sale of animals with deteriorated body condition, and uncommon sales in Northern Burkina Faso [44]. It is possible that destocked herds from previous years affected the availability of animals (and therefore, trade) during 2016; however, favorable rainfall during the second half of 2016 and into 2017 increased pasture availability which could

explain the increase in shipments for 2017 when compared to the previous years. In contrast, the devaluation of the Naira in 2016 (Nigeria's official currency) was unfavorable for livestock exports into Nigeria [45]. Unfavorable conditions because of this devaluation yet trade movements towards Nigeria in 2017 suggest that markets between Niger and Nigeria were included in the data collection sample during 2017. In Mauritania, Tabaski affects spatiotemporal animal movement patterns [21]. Because Tabaski was observed at the end of August, Tabaski-related movements had probably started when our study period finished, which could explain why the number of animal shipments for 2017 are comparable to previous years even when the collection interval was shorter.

We found densely connected market communities that shared more internal links than country-specific communities, and more links than expected by chance. Some of these communities corresponded with trade patterns that were persistent through the years, such the community that moves animals from Central to Southern Côte d'Ivoire and the community within Burkina Faso, while most of them spanned over at least one border, like the groups between Benin and Nigeria, and between Niger and Nigeria. Cross-border trade communities support findings from previous country-specific and regional studies: multiple-country approaches are necessary to support intraregional livestock trade [9,20,23].

Hub markets in border areas and urban centers identified previously in the literature [22,46–48] were confirmed with node volumes and degree centralities. Border markets like Nadiagou, and Dan Barto serve as assembly points where livestock is received from numerous markets located closer to production areas (and therefore have large in-degrees). Urban markets such as Bouaké and export markets like Fada N'Gourma, on the other hand, have a high out-degree centrality and receive long-distance traded livestock before it is shipped to consumer markets. A third of the key markets–(Parakou, Dan Barto and Dèrassi)–were located within 50 km of Nigeria, highlighting the importance of the Nigerian market as a destination for regional livestock production that was not apparent from the country-level findings. These key markets also suggest that Benin and Niger may play an intermediary role in shipments ultimately destined to Nigeria.

Previous efforts to study livestock trade in West Africa have concerned national movements [22], transhumant movements [1] or had limited geographic scope [20,21,23]. We analyzed as a network, for the first time, regional animal movements originating mainly in Burkina Faso, one of the top three live animal exporters in the region. Despite its limitations, our study sheds light on the understudied structure of regional livestock trade and the role of markets in West Africa.

Analysis of existing agricultural product mobility data and its communication to policy makers is essential to advance regional trade integration. Our findings advocate for increasing the density and quality of the regional road network, which could help develop livestock and other intraregional trade products that are primarily transported by vehicle. Border areas should be prioritized as their infrastructure is not prepared to withstand increasing regional trade movements and will likely constrain them [48,49]. Additionally, our results substantiate removing border tariffs and delays that contribute to the higher cost of food in West Africa when compared to other regions of the world [50], and that decrease the population base reachable from border towns [49]. One way forward is to operationalize existing One Stop Border Posts (OSBPs) in important cross-border areas that have been identified in this manuscript.

Some future research directions were identified while carrying out this work. Future efforts to study animal mobility in the region (and its implications for disease surveillance, detection and control) should exploit the bilateral nature of trade by streamlining research initiatives and sharing methods, data, findings and lessons learned. Opportunities also exist to synthesize

knowledge from different sources and fields such as remote sensing products, infrastructure maps, market information systems and household surveys. Given its importance as a consumption market for the region, an effort should be made to better understand live animal shipment patterns into Nigeria. Unfortunately, the current security situation in the Sahel can complicate collecting data in some areas. Finally, using data-informed simulations, previous work has concluded that the risk of regional disease transmission is high in West Africa [20,23]. The risk of regional disease spread can be re-assessed with recent developments in (a) animal mobility patterns in the region and on the (b) inference of network structure from observed network patterns.

## Supporting information

**S1 Table. Percentage of livestock movements by type.**
(DOCX)

**S2 Table. Fitting the degree distribution.**
(DOCX)

**S1 Fig. Proportion of movements by type of movement and month 2013–2017.**
(DOCX)

**S2 Fig. Proportion of movements by type of livestock and month 2013–2017.**
(DOCX)

**S3 Fig. Proportion of movements by type of transport and month 2013–2017.**
(DOCX)

**S1 File. Data collection methodology.**
(DOCX)

**S2 File. Edge list.**
(CSV)

**S3 File. Node list.**
(CSV)

## Acknowledgments

We thank Heather Enloe and Floyid Nicolas for their comments on early versions of this manuscript.

## Author Contributions

**Conceptualization:** Valerie C. Valerio, Olivier J. Walther, Marjatta Eilittä, Rachata Muneepeerakul, Gregory A. Kiker.

**Data curation:** Valerie C. Valerio.

**Formal analysis:** Valerie C. Valerio.

**Funding acquisition:** Gregory A. Kiker.

**Methodology:** Valerie C. Valerio, Olivier J. Walther.

**Resources:** Brahima Cissé.

**Software:** Valerie C. Valerio.

**Supervision:** Olivier J. Walther, Rachata Muneepeerakul, Gregory A. Kiker.

**Visualization:** Valerie C. Valerio.

**Writing – original draft:** Olivier J. Walther, Marjatta Eilittä.

**Writing – review & editing:** Olivier J. Walther, Marjatta Eilittä, Brahima Cissé, Rachata Muneepeerakul, Gregory A. Kiker.

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
