## [Decision Letter · Decision Letter 0]

25 Sep 2019

PONE-D-19-20212

Network analysis of regional livestock trade in West Africa

PLOS ONE

Dear Ms. Valerio,

Thank you for submitting your manuscript to PLOS ONE. After careful consideration, we feel that it has merit but does not fully meet PLOS ONE’s publication criteria as it currently stands. Therefore, we invite you to submit a revised version of the manuscript that addresses the points raised during the review process.

We would appreciate receiving your revised manuscript by Nov 09 2019 11:59PM. To enhance the reproducibility of your results, we recommend that if applicable you deposit your laboratory protocols in protocols.io, where a protocol can be assigned its own identifier (DOI) such that it can be cited independently in the future. For instructions see: http://journals.plos.org/plosone/s/submission-guidelines#loc-laboratory-protocols

We look forward to receiving your revised manuscript.

Kind regards,

Peng Li, Ph.D.

Academic Editor

PLOS ONE

Journal Requirements:

I have read the journal's policy and the authors of this manuscript have the following competing interests: RM serves in the Editorial Board of PLOS ONE.

Reviewers' comments:

Reviewer's Responses to Questions

**Comments to the Author**

1. Is the manuscript technically sound, and do the data support the conclusions?

Reviewer #1: Yes

Reviewer #2: Partly

Reviewer #3: Partly

2. Has the statistical analysis been performed appropriately and rigorously? 

Reviewer #1: Yes

Reviewer #2: Yes

Reviewer #3: Yes

3. Have the authors made all data underlying the findings in their manuscript fully available?

Reviewer #1: No

Reviewer #2: Yes

Reviewer #3: No

4. Is the manuscript presented in an intelligible fashion and written in standard English?

Reviewer #1: Yes

Reviewer #2: Yes

Reviewer #3: Yes

5. Review Comments to the Author

Reviewer #1: General comment

This paper aims to investigate the livestock market and trade network in West Africa. The topic is very interesting and exploratory. A bunch of network analysis metrics is used to discover the pattern of livestock trade origins and destinations. This paper is generally well written. The reviewer recommends minor revision before publishing in this journal. The following comments may help improve the manuscript.

Specific comment:

1. In Line 106-107, the name of the database may need to be explained in English.

2. In Line 265, the p-value is missing.

3. In “Network Analysis Results” and “Discussion and Conclusions” section, please use active voice such as Table 5 shows … instead of referring to the source in parentheses such as …(Table 5).

4. In Line 288, “Large GWCC reinforce” maybe “A large GWCC reinforces”.

5. In Line 111, the authors claim “our findings are then contrasted with existing literature”, please clarify the opinions in the literation and the related conclusion in this paper, and the reason for the difference in the discussion.

Reviewer #2: 1. The significance of the research should be further clarified. Many scholars have studied related content, as listed in the references. In line 35-38, live-animal trade has not been formally studied at a regional scale, you attempt to fill this gap. In fact, it is not a real gap. So where is the innovation of your research?

2. In the paper, one of the conclusion is the trade network has a loose structure (Line 244 and Line 344). What is the criterion for judging whether the structure is loose or tight? Is it described by one indicator or multiple indicators? If multiple indicators are used to describe, an evaluation index system should be given. In addition, compared to what we can say the structure is loose or tight?

3. Why infrastructure construction can affect market structure (Line 351, Line414)? In addition to the infrastructure, what other factors affect the market structure? Please provide evidence.

4. There is too much about the CILSS database (Line107, Line 150, Line330, Line 388). Put the contents about CILSS only in Data source and pre-processing is better.

5. The discussion and conclusions are more likely the results. More specific conclusions need to be reflected. For example, what kind of reginal policies are needed to fulfill the livestock sector’s potential?

Reviewer #3: General comments:

The abstract should be slightly modified to include the weighted property of the studied networks.

Many studies are cited as a basis for the submitted manuscript. An effort has been made at improving the understanding on livestock mobility at regional level, by applying already published methods focusing on the same area, and is appreciated. It will allow consistency in the obtained results and discussion at larger scale. It will also provide support to regional surveillance. However, none explanation nor sufficient details on the used methods are available in the manuscript. The authors should be aware that the reader will have to read all the mentioned published papers to be able to understand the submitted manuscript in a sufficient way... which is not acceptable. Major revision are needed to include few sentences on the "recycle part" of the published studies in order to support the reader understanding.

In addition, non information on the data collection methods and the studied dataset are provided. The authors only refers to a published report supposed to describe both (field data collection method, and related database). Only 2 sentences, provide information on the available data, extracted from the database, and how, the authors cleaned their dataset. Further description have to be provided in the methods section to support the robustness of the used database and the adequacy of the used methods.

Reading the analysis and the shown data, I'm concerned about the 'regional level' appellation... The authors should define the 'region' that they're studying and take into consideration that the geographical region (Fig 1) is not the same that the region concerned by the CILSS database. This database was built by implementing field data collection in 8 country. Niger and Guinea are not part of the CILSS database, but are mentioned as destination and presented on the Fig 1. Benin is part of the CILSS and is not mentioned in the Fig 1 caption. Thanks to the lack of data collection points in Niger and Guinea, the flows could have occurred and being part of the unrecorded data (or recorded in another database). Some discussion on this purpose could be provided.

Moreover, no effort has been made to support a regional understanding on the observed network. General statement are provided as discussion, and deserve further development.

Finally, the writing have to be organised to fit the standard on scientific writing. The current version of the manuscript mixed results and discussion section in multiple part of the manuscript (e.g. L 356, L 361-362).

Other comments:

L 55-56: in the current form, it is unclear about which agricultural products the authors are is talking (species, type of animal products).

L 60-61: "as the projected demand surpasses demographic growth"

If the comment is related to all included at regional scale, the sentence is of poor added value. It'd be valuable to further develop.

L 65: the Livestock trade and livestock products mentioned earlier, are there only related to the trade of meat and live animal in general? Some precisions would be needed to better introduce the study goal. The introduction remain unclear on the purpose: which species, which products, which distinction at regional scale (spatial heterogeneity of the needs and habits related to agriculture...).

L 77-78: the authors should provide precision on what they consider to be source of 'disruptions and shocks'

L 78-81: and related to these livestock data... anthropological studies could be valuable at this scale to better integrate social and human factors and impact on trade habits and agricultural sector. Access to the field is not particularly easy in this area.

L 107 and Material and Methods: The authors should provide sufficient description on the database that they are using in the submitted manuscript (what is CILSS, how the database is organised). The mentioned reference is a 34 pages report which describe a variety of database. Few additional sentences have to be added in order to summarise and support the current study.

Add a brief description on the database, how it was built (compulsory or not, data collection methods).

Is duration of the travel (origin/destination) known?

L 167: The authors have sufficient data to study the market stability at this scale but do not develop. Why the authors decided not to compare the yearly network dataset? Time-series data between 2012-2017 were available but none or a few investigations were done on their structure and evolution over the period. Moreover, even if only 2 trimesters were available in 2017... the recorded data are comparable to those of 2013 and 2016... Do you have information to relate to this point? Could you have part of the explanation?

Discussion on the relation with historical events would be appreciated (2016 = decrease of the national value in Nigeria). And for the regional level? What about quality of the pasture and socio-economical factors (trend in the market, prices... data are part of the CILSS and are from real added value)?

L 207: if ID numbers were collected during the shipment recording... how double counting could have occurred?

L 213: The Tabaski is here mentioned but was not introduced. What is the purpose here? At which scale this religious festivity is practiced? The results are not discussed... The authors should discuss all the results that they are providing, or the results should be withdraw from the submitted manuscript.

Moreover, the Tabaski can hold different name at regional scale and is yearly dependent and occurred between September and October from 2013 to 2017. These specificities could have major impact on the trade habits (related to drought and agricultural/pastoral activities). None discussion of these major aspects was done. An effort should be done to further improve the submitted study and it interest for a regional analysis.

L 332: I'm concerned about the meaning of 'daily data collection'. What is the relation between the recorded date and the effective movement (shipment)? Is this possible that the date of declaration made to the officer do not refers to the date in which the Livestock were traded? In Africa, date can be confusing.

L 352-353: Comparison has to be done carefully. The methods on data collection impact the dataset.

L 393: In the case of Mauritania, the dataset is mostly national with indication on the markets at international scale (regional - origin and destination). The Livestock trade networks were sub-divided into species networks because of the variability on the consumption habits like the Tabaski. At world and regional scale, the authors had highlighted the interest of other socio-cultural habits. Each of the networks were described at both scale, but due to the data collection scheme, it was not possible to implement the predictive study (as well as the finer SNA analysis) at regional scale (end-point markets are not studied, indeed the SNA parameters can't reflect the real centrality of these markets). The methods from Apolloni et al. and Nicolas et al. provided robustness and consistency to the analysis regarding the input dataset (field data collection was scheduled for the purpose of the Livestock trade network analysis). The level of detail is finer, however, only one year was available!

Comparison between the two studies has to be done carefully...

L 406-408: Requirement of a standardised data collection methods at regional scale, would also be expected

S7 Fig. Give the full country name in the figure caption as in the Fig 1.

6. PLOS authors have the option to publish the peer review history of their article (what does this mean?). If published, this will include your full peer review and any attached files.

Reviewer #1: No

Reviewer #2: No

Reviewer #3: No

---

## [Author Response · Author response to Decision Letter 0]

17 Feb 2020

Response to reviewers: letter responding to each point raised by the academic editor and reviewer(s)

Dear Dr. Peng Li, 

Many thanks for your excellent review of our submission. We have carefully reviewed and responded to all the points raised by yourself and the three reviewers. Please find each point raised and its response below in its corresponding section: Academic Editor, Reviewer #1, Reviewer #2 and Reviewer #3. Specific comments within each section were numbered as E1, E2 (editor section), R1.1, R2.2 (reviewer 1 section), and so on, for simplicity. We have formatted the response to each general and specific comment in red text.

Academic Editor 

Journal Requirements:

E1. When submitting your revision, we need you to address these additional requirements.

RESPONSE: 

We appreciate that the Editor called this to our attention. We have made the following changes to meet PLOS ONE’s style requirements and to address reviewer #2 comments (R2.5):

• Modified the style of “Introduction” to heading level 1 (bold, 18pt)

• Corrected the level 3 heading (Network Analysis) to sentence case (changed to Network analysis in bold, 14pt)

• Revised level 2 heading “Network Analysis” to sentence case (“Network analysis”)

• Changed level 1 heading “Discussion and Conclusions” to “Discussion”

• Separated the Supplementary Information file into individual files named according to PLOS ONE’s guidelines (SI_1, SI_2, …).

E2. Thank you for stating the following in the Competing Interests section:

I have read the journal's policy and the authors of this manuscript have the following competing interests: RM serves in the Editorial Board of PLOS ONE.

RESPONSE: 

We have updated our cover letter, adding a competing interests statement that now reads:

“RM (co-author) serves in the Editorial Board of PLOS ONE. However, this does not alter our adherence to PLOS ONE policies on sharing data and materials.”

We also included a data availability statement that now reads:

“Data are available as a supporting information file of the manuscript (S7-8).”

E3. We note that you have indicated that data from this study are available upon request. PLOS only allows data to be available upon request if there are legal or ethical restrictions on sharing data publicly. For more information on unacceptable data access restrictions, please see http://journals.plos.org/plosone/s/data-availability#loc-unacceptable-data-access-restrictions.

RESPONSE: 

We have included the data as a supporting information file of the manuscript. Our cover letter has been modified to reflect this:

“Data are available as a supporting information files in the manuscript (S7-8).”

Reviewer # 1

R1 General comment

This paper aims to investigate the livestock market and trade network in West Africa. The topic is very interesting and exploratory. A bunch of network analysis metrics is used to discover the pattern of livestock trade origins and destinations. This paper is generally well written. The reviewer recommends minor revision before publishing in this journal. The following comments may help improve the manuscript.

RESPONSE: 

We thank Reviewer 1 for their time and comments. Below we address each specific point raised, in order.

R1 Specific comments

R1.1. In Line 106-107, the name of the database may need to be explained in English.

RESPONSE: 

Lines 100-101 have been modified to include the name of the database in English:

“Permanent Interstate Committee for Drought Control in the Sahel (Comité Inter-Etats pour la Lutte contre la Sécheresse au Sahel, CILSS).”

R1.2. In Line 265, the p-value is missing.

RESPONSE: 

All the p-values have been included (now L315-317). The reader is also referred to S5 table for all the details. Thank you. 

R1.3. In “Network Analysis Results” and “Discussion and Conclusions” section, please use active voice such as Table 5 shows … instead of referring to the source in parentheses such as …(Table 5).

RESPONSE: 

We have modified the Results (L228-358) and Discussion (L386-484) sections to the active voice.

R1.4. In Line 288, “Large GWCC reinforce” maybe “A large GWCC reinforces”.

RESPONSE: 

This sentence is no longer included in the manuscript. We thank the reviewer for noticing this error.

R1.5. In Line 111, the authors claim “our findings are then contrasted with existing literature”, please clarify the opinions in the literation and the related conclusion in this paper, and the reason for the difference in the discussion.

RESPONSE: We appreciate the reviewer’s suggestion. To address this comment and reviewers # 2 and # 3 concerns on manuscript organization and clarity, we have done the following:

• Separated and Rewrote the Results and Discussion sections

• Included possible reasons for differences between our findings and existing studies in the Discussion (L372-385)

Reviewer # 2

R2 Specific comments

R2.1. The significance of the research should be further clarified. Many scholars have studied related content, as listed in the references. In line 35-38, live-animal trade has not been formally studied at a regional scale, you attempt to fill this gap. In fact, it is not a real gap. So where is the innovation of your research?

RESPONSE: 

We have reframed the importance of our work in the Introduction (L93-99) as follows:

“Although some understanding of animal mobility in the region exists, trade patterns of animals originating in the top exporters (Niger, Mali and Burkina Faso) have not been quantitatively analyzed as a network.

Our objective is to advance the formal study of regional trade networks in West Africa. We use network analysis to characterize the trade network of live animals (cattle, sheep, goats and donkeys) originating primarily in Burkina Faso, the third largest exporter of live animals in the region [9] as they are shipped through the central and eastern trade basins.”

We agree that some scholars have covered related content before in West Africa (Lines 82-95); however, previous work lacks the geographic and time breadth that the data used in our study has. Of the previous 5 formal (quantitative) network analyses of animal mobility in West Africa:

• The study by Thébaud et al [1] concerned transhumant movements only [1], differing from ours because, although some animals can be sold during transhumance, seasonal movements are not primarily fueled by trade. Additionally, this study did not analyze animal movements as a network.

• Apolloni et al. [2] described preliminary findings from a survey on animal movements conducted in Mauritania. Nicolas et al. [3] then attempted to predict these movements within Mauritania only, even though 70% of the animals in their database had crossed borders. Because their work was focused on animals entering and leaving Mauritania, it differs from ours.

• Two studies investigated the spread of diseases regionally through animal movements: Dean et al. [4] and Motta et al. [5]. Dean et al. [4] evaluate the risk of disease spread from Togo to neighboring countries, and from Burkina Faso into Togo. Data was collected in the Savannah region of northern Togo only; therefore, this study differs fundamentally from ours in both its objective (evaluating disease transmission risk) and its geographic extent. It is worth mentioning that CILSS’s database does capture animals entering Togo from Burkina Faso in comparable numbers as the results presented by Dean et al (~40,000 heads in Jan-August 2017) but does not capture animals leaving Togo as well as their study. On the other hand, Motta et al. [5] do not explicitly perform simulations of disease transmission risk, but describe the structure of the Cameroonian animal movement network, discussing the implications for disease spread. This study differs from ours primarily in the geographic coverage of the movement network, although some of the objectives are similar.

Our manuscript focuses on regional movements of animals that originate in Burkina Faso, one of the top three live animal exporters in West Africa (according to the World Bank [6]), and therefore complements the existing literature. Furthermore, findings at three scales of analysis are combined to make sense of regional patterns of trade and identify important locations for the regional network. 

Although some studies have covered this subject in the past, we believe that regional trade of live animals in West Africa is still quantitatively understudied. Studying animal trade patterns is important because live animals are the most important food product traded in the region, official statistics do not adequately capture their flows, and because describing trade patterns can inform regional policies on trade, disease control and food security.

R2.2. In the paper, one of the conclusions is the trade network has a loose structure (Line 244 and Line 344). What is the criterion for judging whether the structure is loose or tight? Is it described by one indicator or multiple indicators? If multiple indicators are used to describe, an evaluation index system should be given. In addition, compared to what we can say the structure is loose or tight?

RESPONSE: 

We have removed this line from the paragraph. Instead, we compare the observed networks to their corresponding ensemble generated with a null (configuration) model that maintains the observed degree sequence (L 293, 301). With these simulated networks, we calculated a z score for each observed network metric value. Some metrics had to be excluded because they could not be calculated. The density is one of the metrics that could not be benchmarked against the simulated networks, because by construction all the networks in the configuration model ensemble had the same density. We thank the reviewer for this very useful comment.

R2.3. Why infrastructure construction can affect market structure (Line 351, Line414)? In addition to the infrastructure, what other factors affect the market structure? Please provide evidence.

RESPONSE:

Infrastructure can affect the observed market structure because survey points were located on major trade corridors. These trade corridors rely on transnational paved roads that help connect the Sahel to consumption centers [7]. Therefore, it is possible that the location of the survey points affects the observed trade network structure. Most movement being made by vehicle (in contrast to findings in other West African countries) support this claim. We moved those lines to L413-417, that now read:

“Our findings reflect the fragmentation of the road network inherited from the colonial period and the poor accessibility of many peripheral markets in the region. Cattle trade relies on a handful of paved roads in each country and on a limited number of transnational routes that help connect the Sahel to the main consumption centers [49].”

In a broader sense, infrastructure can affect the market structure by incentivizing development and inflow of people and goods towards newly accessible locations. For example, the construction of the Nouakchott-Nouadhibou route in Mauritania in 2014 incentivized fishermen and herdsmen to relocate near the paved road to access new markets [8]. In the case of livestock, historical trade routes inherited from the colonial period have conditioned how and where live animals are traded (in the form of ethnic and social ties, and a fragmented road network).

R2.4. There is too much about the CILSS database (Line107, Line 150, Line330, Line 388). Put the contents about CILSS only in Data source and pre-processing is better.

RESPONSE: 

We have moved details on the database to the Data source and pre-processing sub-section. Because no scientific publication exists that describes this database in detail, we included a supplementary information file (S6 File. Data collection methodology) with more information about the data collection and also referred the reader to the official documentation of CILSS. 

R2.5. The discussion and conclusions are more likely the results. More specific conclusions need to be reflected. For example, what kind of regional policies are needed to fulfill the livestock sector’s potential?

RESPONSE: 

To address this and other reviewer comments, we have modified the sections as follows:

• “Network Analysis Results” has been renamed “Results”. The content has been modified accordingly

• The “Discussion and conclusions” section has been renamed “Discussion” 

In the conclusions, we have recommended the following actions concerning regional policies (L463-472):

• Increasing the density and quality of the regional road network to support trade of animals and other food products that are primarily transported by vehicle, prioritizing cross-border areas important for regional trade

• Removing border delays and costs that contribute to the higher cost of food in West Africa when compared to other regions of the world. One specific way forward is to (install) operationalize One Stop Border Posts (OSBPs) in known cross-border areas of importance (some of which have been identified in the manuscript)

• Streamlining regional animal movement and disease surveillance/control research: sharing findings, best practices and lessons learned. This will enable synergies and avoid duplication of efforts led by different countries and/or international organizations

Additionally, the following research priorities have been identified (L473-484):

• Studying spatiotemporal patterns of animal flows into Nigeria (an important regional consumption market)

• Assessing the risk of disease spread at a regional scale using recent developments in (a) regional patterns of animal mobility, (b) inference of network structure from observed trade patterns (see [9]) 

Reviewer #3

R3 General comment:

The abstract should be slightly modified to include the weighted property of the studied networks.

Many studies are cited as a basis for the submitted manuscript. An effort has been made at improving the understanding on livestock mobility at regional level, by applying already published methods focusing on the same area, and is appreciated. It will allow consistency in the obtained results and discussion at larger scale. It will also provide support to regional surveillance. However, none explanation nor sufficient details on the used methods are available in the manuscript. The authors should be aware that the reader will have to read all the mentioned published papers to be able to understand the submitted manuscript in a sufficient way... which is not acceptable. Major revision are needed to include few sentences on the "recycle part" of the published studies in order to support the reader understanding.

In addition, non information on the data collection methods and the studied dataset are provided. The authors only refers to a published report supposed to describe both (field data collection method, and related database). Only 2 sentences, provide information on the available data, extracted from the database, and how, the authors cleaned their dataset. Further description have to be provided in the methods section to support the robustness of the used database and the adequacy of the used methods.

Reading the analysis and the shown data, I'm concerned about the 'regional level' appellation... The authors should define the 'region' that they're studying and take into consideration that the geographical region (Fig 1) is not the same that the region concerned by the CILSS database. This database was built by implementing field data collection in 8 country. Niger and Guinea are not part of the CILSS database, but are mentioned as destination and presented on the Fig 1. Benin is part of the CILSS and is not mentioned in the Fig 1 caption. Thanks to the lack of data collection points in Niger and Guinea, the flows could have occurred and being part of the unrecorded data (or recorded in another database). Some discussion on this purpose could be provided.

Moreover, no effort has been made to support a regional understanding on the observed network. General statement are provided as discussion, and deserve further development.

Finally, the writing have to be organised to fit the standard on scientific writing. The current version of the manuscript mixed results and discussion section in multiple part of the manuscript (e.g. L 356, L 361-362).

RESPONSE: 

We thank Reviewer 3 for their time and comments. Below we address each point raised in the general commentary, in order.

The abstract has been modified to reflect the weighted nature of the networks (L30).

A summary of previous related work and how it is similar or differs from our manuscript has been included in the introduction (L83-96). This paragraph summarizes previous work and how it relates and differs from this manuscript.

We have included more information on the methods. The data collection methodology is described in S6 File in enough detail for the reader to understand how the data was collected, processed and stored. The reader is referred to the website that contains all the official documentation of the data collection (www.agrictrade.net) for more information. At the time this revision was written, this website was being finalized. The data necessary to replicate our analysis are provided in S7-S8 Files. Lines 130-170 detail the steps taken to pre-process the database before analysis, while S7-S8 Files suffice to rebuild the yearly networks. The variables in the provided data are described in Table 1. Concerning the “recycled” part of the paper, the fourth paragraph in the introduction (L82-95) now briefly explains the livestock network literature for West Africa and explains how our study differs from/is similar to it.

The significance and geographic scope of our work has been updated. Although we intended to include more countries in our analysis, many entries in the database were incomplete and therefore could not be included. Our final dataset comprises movements originating primarily in Burkina Faso, one of the top three exporter of animals in the region. The data tracks these animals as they are transported primarily in a North-South direction into Benin, Cote d’Ivoire, Ghana and Nigeria. Thus, our study region comprises the Central trade basin and part of the Eastern trade basin of West Africa (this has been stated clearly in the manuscript, L98, L138-142). 

The manuscript has been reorganized for clarity, as suggested by the reviewer. We have modified the content of each section so that:

• The “Results” section only presents our findings

• The “Discussion” section interprets and contextualizes the results 

R3 Specific comments

R3.1 L 55-56: in the current form, it is unclear about which agricultural products the authors are is talking (species, type of animal products).

RESPONSE: 

L 55-56 now reads: “…therefore unknown. Livestock (mostly cattle and small ruminants) are traded live and constitute the leading product in food trade for the region [8,9]” 

R3.2 L 60-61: "as the projected demand surpasses demographic growth"

If the comment is related to all included at regional scale, the sentence is of poor added value. It'd be valuable to further develop.

RESPONSE:

We have removed the sentence and instead added:

“…the existing gap between production and demand of animal products is expected to widen in the next decades driven by population and income growth, migration and urbanization“ (L59-61)

R3.3 L 65: the Livestock trade and livestock products mentioned earlier, are there only related to the trade of meat and live animal in general? Some precisions would be needed to better introduce the study goal. The introduction remain unclear on the purpose: which species, which products, which distinction at regional scale (spatial heterogeneity of the needs and habits related to agriculture...).

RESPONSE:

We have modified this paragraph. We have specified the product and species (live cattle and small ruminants, L56), and summarized it again in the last paragraph of the introduction where the objectives are stated (L98-99).

R3.4 L 77-78: the authors should provide precision on what they consider to be source of 'disruptions and shocks'

RESPONSE:

We have identified two instances of disruptions and/or shocks (climatic and conflict events, L77-78). These types of disruptions can affect both the production of animals and the functioning of markets (and therefore trade). 

R3.5 L 78-81: and related to these livestock data... anthropological studies could be valuable at this scale to better integrate social and human factors and impact on trade habits and agricultural sector. Access to the field is not particularly easy in this area.

RESPONSE:

We agree with the reviewer. Much could be done with other types of studies. We believe that spaces exist for synergies between different types of studies to better understand agricultural trade patterns (L478). 

R3.6 L 107 and Material and Methods: The authors should provide sufficient description on the database that they are using in the submitted manuscript (what is CILSS, how the database is organised). The mentioned reference is a 34 pages report which describe a variety of database. Few additional sentences have to be added in order to summarise and support the current study.

Add a brief description on the database, how it was built (compulsory or not, data collection methods).

RESPONSE:

We thank the reviewer for highlighting this. We have added “S6 File. Data collection methodology” with more details on the CILSS and describing their data collection purpose and methodology. Additionally, Table 1 describes each variable present in the data provided in S7 File and S8 File. We have removed the report from the references and instead cited more appropriate sources. The data collection was not compulsory for traders, but to our knowledge no problems were reported with trader participation.

R3.7 Is duration of the travel (origin/destination) known?

RESPONSE:

The duration of travel is unknown.

R3.8 L 167: The authors have sufficient data to study the market stability at this scale but do not develop. Why the authors decided not to compare the yearly network dataset? Time-series data between 2012-2017 were available but none or a few investigations were done on their structure and evolution over the period. Moreover, even if only 2 trimesters were available in 2017... the recorded data are comparable to those of 2013 and 2016... Do you have information to relate to this point? Could you have part of the explanation?

RESPONSE:

We briefly present the evolution of the network in Table 5 in the form of network statistics for each year. In addition, Fig 5 presents each yearly network. We did not elaborate on market stability for the following reasons:

• Missing data: As is reported in the manuscript, many movements are missing either the origin or destination market. Only complete, geolocated entries were used in our analysis.

• Changes in data collection: The number of markets surveyed changed between 2013-2017 (L287). Throughout the study period, survey points have been added and removed from the sample. 

We chose to not study the evolution of the market structure in detail, because changes can be an artifact of the data collection and not reflect true variation in the trade network.

Regarding the comparability of 2017 (Q1-Q3) volumes with previous years, the following can explain this (L428):

• More markets were surveyed in 2017 than in previous years 

• FEWSNET reported above-average cumulative seasonal rainfall for Burkina Faso in July 2017; At the end of 2017 livestock were reported to be in good physical condition, which suggests that enough biomass was available during a number of months prior 

• Tabaski is set on the lunar calendar and its date for 2017 was at the end of August; significantly more sheep movements had been reported by August in 2017 than for previous years, when Tabaski was later

• When the duration of the data collection for each year is considered, more movements were made in 2014 than in 2017 (2017 was a “normal” production year according to FEWSNET, https://fews.net/west-africa/burkina-faso/key-message-update/july-2017 )

R3.9 Discussion on the relation with historical events would be appreciated (2016 = decrease of the national value in Nigeria). And for the regional level? What about quality of the pasture and socio-economical factors (trend in the market, prices... data are part of the CILSS and are from real added value)?

RESPONSE: 

The naira devaluation and other climatic conditions are now discussed in L417-437. In the discussion, the “Limitations” subsection discusses further why observed changes can be reflecting changes in the data collection.

Regarding market price trends, FEWSNET did not publish livestock prices in their price bulletin series for Burkina Faso. We did not find other official sources of livestock market prices elsewhere, so we could not use them.

R3.10 L 207: if ID numbers were collected during the shipment recording... how double counting could have occurred?

RESPONSE: In this manuscript, we refrain from reporting on total volumes because of possible double counting of animals (L361). CILSS’s objective is to track the magnitude, direction and changes in intraregional trade, and they aim to capture both official and informal movements. Therefore, animal/official custom transaction IDs were not collected during the study period. Under their collection methodology (see S6 File for details), we presume that multiple counting could have happen in three ways: 

• An animal is counted early in the trade journey and then counted in another section of the journey. For example, a truck shipment with cattle is registered at Bobo-Dioulasso (collection point in Burkina Faso) as leaving Bobo-Dioulasso towards Daloa (in Cote d’Ivoire). There is a possibility that the same truck shipment is registered at Man (collection point in Cote d’Ivoire) point as going to Abidjan (also in Cote d’Ivoire). 

• An animal is counted as part of a specific shipment in a specific data collection pint. Later, this animal leaves the initial shipment it was counted on and becomes part of a new shipment (for example, it is sold to another trader). This new shipment is later reported at another survey point.

• The same shipment is counted by each of the two enumerators at the collection point and reported to the focal point twice. The data collection methodology has measures in place to prevent this type of double counting (enumerators compare and collate their records at the end of each survey day, see S6 File). 

If we add all the shipments to report on total volumes, we run the risk of counting some animals more than once and have no reliable method to detect these duplicates.

R3.11 L 213: The Tabaski is here mentioned but was not introduced. What is the purpose here? At which scale this religious festivity is practiced? The results are not discussed... The authors should discuss all the results that they are providing, or the results should be withdraw from the submitted manuscript.

Moreover, the Tabaski can hold different name at regional scale and is yearly dependent and occurred between September and October from 2013 to 2017. These specificities could have major impact on the trade habits (related to drought and agricultural/pastoral activities). None discussion of these major aspects was done. An effort should be done to further improve the submitted study and it interest for a regional analysis.

RESPONSE:

We have now introduced Tabaski to the reader in the second paragraph of the results (L235). We specified that it is not observed on the Gregorian calendar and therefore is celebrated in a different date each year. The importance of the holiday for the region and its consumption patterns is highlighted (), and Alternative names for Tabaski were also introduced to the reader (L234). We have also included a paragraph discussing the possible climatic, historical and data collection conditions that could explain the temporal changes in our networks. (L 417-437)

R3.12 L 332: I'm concerned about the meaning of 'daily data collection'. What is the relation between the recorded date and the effective movement (shipment)? Is this possible that the date of declaration made to the officer do not refers to the date in which the Livestock were traded? In Africa, date can be confusing.

RESPONSE:

We appreciate that the reviewer has brought up this point of concern. CILSS’s data collection is carried out daily or on market days in selected markets and border-crossing points in trade corridors (see S6 File for more details). On data collection days, two enumerators visit each survey point. Movements are recorded by both enumerators on the day that they take place, and not when they are reported to official authorities or when they are collated and transferred to the focal point in charge of each country-commodity. This is reflected in the dates reported in the data (not shown in the manuscript because of the time scale of the analysis). Furthermore, movement data collected in each survey point is transferred to a focal point at the end of each month. Therefore, if the date of the movement is estimated, which is highly unlikely given the collection methodology, collation of data by month to the focal point and subsequent transfer to the CILSS (and a yearly time step in the manuscript) would lessen the impact of day-differences between the actual movement date and estimated dates. Furthermore, the ASSESS project (USAID/West Africa Analytical Support Services and Evaluations for Sustainable Systems in Agriculture, Environment, and Trade project) carried out a Data Quality Assessment (DQA) of CILSS’s intra-regional trade monitoring in 2017. Although the DQA found some improvement opportunities, no issues concerning discrepancies on movement dates were raised. We understand that data collection in the field is far from perfect and appreciate this concern.

R3.13 L 352-353: Comparison has to be done carefully. The methods on data collection impact the dataset.

RESPONSE: 

We agree that comparison should be handled carefully. We have all direct comparison between the networks and instead compared the observed network metrics to the ensemble of networks generated with the configuration model (L293). Direct comparisons between the studies are not made with the caveat that the data collection efforts were different (L372). Although it is possible to compare different networks with novel graph distance and other types of algorithms, that is outside of our manuscript scope.

R3.14 L 393: In the case of Mauritania, the dataset is mostly national with indication on the markets at international scale (regional - origin and destination). The Livestock trade networks were sub-divided into species networks because of the variability on the consumption habits like the Tabaski. At world and regional scale, the authors had highlighted the interest of other socio-cultural habits. Each of the networks were described at both scale, but due to the data collection scheme, it was not possible to implement the predictive study (as well as the finer SNA analysis) at regional scale (end-point markets are not studied, indeed the SNA parameters can't reflect the real centrality of these markets). The methods from Apolloni et al. and Nicolas et al. provided robustness and consistency to the analysis regarding the input dataset (field data collection was scheduled for the purpose of the Livestock trade network analysis). The level of detail is finer, however, only one year was available!

Comparison between the two studies has to be done carefully...

RESPONSE: 

We agree with the reviewer. We have removed direct comparisons between the network structures and highlight why different network should not be compared in the manuscript (L372).

R3.15 L 406-408: Requirement of a standardised data collection methods at regional scale, would also be expected

RESPONSE: 

We thank the reviewer for highlighting the need for more information concerning the data collection. The data collection was coordinated by the CILSS and carried out with the support of 10 regional organizations involved in trade. Standardized methods were used in the data collection (in the form of enumerator training and standardized collection forms), and throughout the data pipeline (from collection, collation, storage, entry, transfer, analysis to reporting). We have included “S6 File. Data collection methodology” in the Supplementary Information to provide more details on the data collection methods.

R3.16 S7 Fig. Give the full country name in the figure caption as in the Fig 1.

RESPONSE: S7 Fig has been replaced by S4 Fig and the ISO codes are no longer necessary.

References

1. Thébaud B. Pastoral and Agropastoral Resilience in the Sahel: Portrait of the 2014-2015 and 2015-2016 Transhumance [Internet]. 2017. Available from: http://www.inter-reseaux.org/IMG/pdf/afl_resilience_study_june2017_abridged_version.pdf

2. Apolloni A, Nicolas G, Coste C, El Mamy AB, Yahya B, El Arbi AS, et al. Towards the description of livestock mobility in Sahelian Africa: Some results from a survey in Mauritania. PLoS One. 2018;13(1):1–24. 

3. Nicolas G, Apolloni A, Coste C, Wint GRW, Lancelot R, Gilbert M. Predictive gravity models of livestock mobility in Mauritania: The effects of supply, demand and cultural factors. PLoS One. 2018;13(7):1–21. 

4. Dean AS, Fournié G, Kulo AE, Boukaya GA, Schelling E, Bonfoh B. Potential Risk of Regional Disease Spread in West Africa through Cross-Border Cattle Trade. PLoS One. 2013;8(10):1–9. 

5. Motta P, Porphyre T, Handel I, Hamman SM, Ngu Ngwa V, Tanya V, et al. Implications of the cattle trade network in Cameroon for regional disease prevention and control. Sci Rep [Internet]. 2017 Apr 7;7(1):43932. Available from: http://www.nature.com/articles/srep43932

6. World Bank. Connecting Food Staples and Input Markets in West Africa - a Regional Trade Agenda for ECOWAS Countries [Internet]. Washington, DC; 2015. Report No.: 97279-AFR. Available from: https://openknowledge.worldbank.org/handle/10986/22276

7. OECD/SWAC. Accessibility and Infrastructure in Border Cities. Paris: OECD Publishing; 2019. (West African Papers). 

8. Steck B. West Africa facing the lack of traffic lanes. EchoGéo 20. 2012; 

9. Chaters GL, Johnson PCD, Cleaveland S, Crispell J, de Glanville WA, Doherty T, et al. Analysing livestock network data for infectious disease control: an argument for routine data collection in emerging economies. Philos Trans R Soc B Biol Sci [Internet]. Royal Society; 2019 Jul 8;374(1776):20180264. Available from: https://doi.org/10.1098/rstb.2018.0264

---

## [Decision Letter · Decision Letter 1]

21 Apr 2020

Network analysis of regional livestock trade in West Africa

PONE-D-19-20212R1

Dear Dr. Valerio,

We are pleased to inform you that your manuscript has been judged scientifically suitable for publication and will be formally accepted for publication once it complies with all outstanding technical requirements.

With kind regards,

Peng Li, Ph.D.

Academic Editor

PLOS ONE

Additional Editor Comments (optional):

Reviewers' comments:

Reviewer's Responses to Questions

**Comments to the Author**

1. If the authors have adequately addressed your comments raised in a previous round of review and you feel that this manuscript is now acceptable for publication, you may indicate that here to bypass the “Comments to the Author” section, enter your conflict of interest statement in the “Confidential to Editor” section, and submit your "Accept" recommendation.

Reviewer #1: All comments have been addressed

Reviewer #2: All comments have been addressed

2. Is the manuscript technically sound, and do the data support the conclusions?

Reviewer #1: Yes

Reviewer #2: Yes

3. Has the statistical analysis been performed appropriately and rigorously? 

Reviewer #1: Yes

Reviewer #2: Yes

4. Have the authors made all data underlying the findings in their manuscript fully available?

Reviewer #1: Yes

Reviewer #2: Yes

5. Is the manuscript presented in an intelligible fashion and written in standard English?

Reviewer #1: Yes

Reviewer #2: Yes

6. Review Comments to the Author

Reviewer #1: The reviewer does not have further comments and recommends this manuscript to be published in this journal.

Reviewer #2: (No Response)

7. PLOS authors have the option to publish the peer review history of their article (what does this mean?). If published, this will include your full peer review and any attached files.

Reviewer #1: No

Reviewer #2: No

---

## [Editor Report · Acceptance letter]

29 Apr 2020

PONE-D-19-20212R1 

Network analysis of regional livestock trade in West Africa 

Dear Dr. Valerio:

I am pleased to inform you that your manuscript has been deemed suitable for publication in PLOS ONE. Congratulations! Your manuscript is now with our production department. 

With kind regards,

on behalf of

Dr. Peng Li 

Academic Editor

PLOS ONE